# Prenatal Programming of Neuroendocrine System Development by Lipopolysaccharide: Long-Term Effects

**DOI:** 10.3390/ijms19113695

**Published:** 2018-11-21

**Authors:** Marina Izvolskaia, Viktoria Sharova, Liudmila Zakharova

**Affiliations:** Koltsov Institute of Developmental Biology, Russian Academy of Sciences, 119334 Moscow, Russia; izvolskaya@hotmail.com (M.I.); l-a-zakharova@mail.ru (L.Z.)

**Keywords:** lipopolysaccharide, GnRH neurons, HPG system, cytokines, critical periods of development

## Abstract

Various stress factors during critical periods of fetal development modulate the epigenetic mechanisms controlling specific genes, which can affect the structure and function of physiological systems. Maternal immune stress by bacterial infection simulated by lipopolysaccharide (LPS) in an experiment is considered to be a powerful programming factor of fetal development. Studies of the molecular mechanisms controlling the formation and functioning of physiological systems are in the pilot stage. LPSs are the most potent natural inflammation factors. LPS-induced increases in fetal levels of pro- and anti-inflammatory cytokines can affect brain development and have long-term effects on behavior and neuroendocrine functions. The degradation of serotonergic neurons induced by LPS in the fetus is attributed to the increased levels of interleukin (IL)-6 and tumor necrosis factor (TNFα) as well as to anxiety and depression in children. Dopamine deficiency causes dysthymia, learning disability, and Parkinson’s disease. According to our data, an LPS-induced increase in the levels of IL-6, leukemia inhibitory factor (LIF), and monocyte chemotactic protein (MCP-1) in maternal and fetal rats during early pregnancy disturbs the development and functioning of gonadotropin-releasing hormone production and reproductive systems. It is important to note the high responsiveness of epigenetic developmental mechanisms to many regulatory factors, which offers opportunities to correct the defects.

## 1. Introduction

An increasing body of experimental and clinical data indicates a negative effect of various stress factors including viral and bacterial infection on the development of fetal physiological systems. Unfavorable impacts on the fetus during critical ontogenetic periods can affect the molecular mechanisms controlling the formation and functioning of these systems. Their plasticity allows adaptation to changes even in early ontogeny. At the same time, the induced modifications can disturb homeostatic systems that underpin the adult body adaptation and increase the risk of various pathologies in offspring [1,2,3,4,5].

Experimental studies often use lipopolysaccharide (LPS), a major component of the outer membrane in gram-negative bacteria and enterobacteria in particular, to mimic a bacterial infection without harmful consequences. LPS is one of the most potent natural inducers of inflammation coupled with increased synthesis of cytokines, enzymes, eicosanoids, adhesive molecules, and free radical formation. It is generally accepted that LPS activates the maternal immune system, which enhances the synthesis of pro- and anti-inflammatory cytokines in both maternal and fetal organisms. Consequently, cytokines promote the secretion of a hormonal cascade in the hypothalamic–pituitary–adrenal system (HPA), thus eliciting the hormonal response to stress [4,6,7].

In this review, we analyze original and published data on possible mechanisms underlying the effect of *Escherichia coli* LPS on the development of the HPA and hypothalamic–pituitary–gonadal (HPG) systems and their functioning in adults.

## 2. LPS Structure and Induction of Cytokine Synthesis

Lipopolysaccharides are high-molecular substances combining both hydrophilic (carbohydrate) and hydrophobic (lipid) parts within the same molecule [8]. LPSs are composed of three covalently bound components: A conserved biphosphorylated lipid A, a central core, and a polysaccharide O antigen [8,9,10]. After bacterial lysis, lipid A and the core are released into the blood, which can cause intoxication and septic shock. The polysaccharide component is the most variable part of the molecule. It is recognized by the host’s immune system and its structure (O antigens) determines the specificity of the immune response to bacterial strains [8]. Depending on the length and variations in the polysaccharide part, the molecular weight of the monomeric LPS ranges from 10 to 20 kDa. Due to the hydrophilic polysaccharide component, LPSs are water-soluble yielding stable micellar aggregates. However, such aggregates induce no pronounced cell response [9,11]. LPS complexes with proteins in the outer cell wall of gram-negative bacteria form complexes called bacterial endotoxins. This term applies to both natural endotoxins present in the bacterial lysate and to peptide-free LPS preparations.

After intravenous or intraperitoneal administration, LPS largely accumulates in the liver, as well as in the spleen and walls of blood vessels and air bladders. In antigen-presenting cells and macrophages in particular, a complex of LPS with CD14 and MD2 receptors triggers the danger signal via the Toll-like receptors—TLR4 [12]. This initiates the cascade of intracellular responses involving various kinases releasing the transcription factor NF-kB from the complex with IkB kinase. NF-kB enters the nucleus and binds the promoters of proinflammatory cytokines and costimulatory proteins [13].

## 3. LPS Effect on Permeability of Blood–Brain, Blood–Testis, and Blood–Placenta Barriers

Despite the common mechanisms of cytokine synthesis within the body, several immune-privileged organs protected from immune reactions by specific blood barriers after activation of the immune system are recognized. Such organs include the brain, eye, testis, and uterus with a developing fetus [14]. LPS transport across these barriers is usually complicated and depends on the structure and origin of the molecule, while LPS effects partially depend on the synthesis and secretion of pro- and anti-inflammatory cytokines by the structures forming these barriers [15].

The adult brain is protected from LPS by the blood–brain barrier (BBB). It is undetectable in the animal brain after intravenous administration of radiolabeled LPS [16]. At the same time, endothelial cells forming the BBB have sites binding LPS and its complex with accessory proteins [17]. LPS induces the synthesis of interleukins (IL)-1α, IL-1β, IL-6, and IL-10, tumor necrosis factor (TNFα), granulocyte–macrophage colony-stimulating factor (GM-CSF), and leukemia inhibitory factor (LIF) [18,19]. The accompanying inflammation improves the signal passage via vascular endothelial cells and the transport of cytokines across the BBB. Proinflammatory cytokines circulating in the peripheral blood, in particular LIF, which are upregulated by LPS can also enter the brain using the special transport system [19]. The effect of LPS on the brain can be mediated by TLR4 activation independently of the systemic triggering of cytokines. TLR4 transcription in the mouse brain has been demonstrated in the meninges, ventricular ependymal, circumventricular organs, and microglia [20,21].

Only fragmentary data are available on LPS transport across the blood–testis barrier. A recent study has demonstrated that intraperitoneal administration of LPS to adult male mice increases the transport of a fluorescent tag (fluorescein isothiocyanate isomer) across the barrier and decreases the production of the membrane protein occludin involved in tight junction formation between Sertoli cells, which is required for barrier establishment. In addition, Sertoli cells demonstrated an increased level of mitogen-activated protein kinase phosphatase (MKP)-1, which attenuated the negative effects of LPS through the interaction with p38 MAP kinase and IκBα molecules [22].

The blood–placenta barrier is of particular significance for the developing organism and barrier permeability is similar in human and rodents for various molecules [23]. The fetus and placenta are closely connected to the maternal circulatory system, which allows the humoral factors to migrate between the tissues. LPS transport across the barrier remains a subject to debate. Peripheral administration of LPS from gram-negative bacteria is a well-characterized model of inflammation in rodents [4]. A single intraperitoneal injection of a low doses (50–100 µg/kg) of 125I-labeled LPS to pregnant mice has revealed no LPS in fetal tissues [24], while a radioactive signal was detected in both placental and fetal tissues after an intravenous injection of a higher dose (up to 1000 µg/kg) [1].

## 4. Role of Cytokines in Mother–Fetus Interaction and Induction of Their Synthesis by LPS

Simamura et al. [25] proposed a hypothetical model of the mother–fetus interaction mediated by the pleiotropic cytokine LIF in rats. According to this model, LIF administered to a female on day 15.5 of pregnancy stimulated the secretion of adrenocorticotropic hormone (ACTH) in both the placenta and fetus. ACTH binds melanocortin receptor types 2 and 5 (MC2/5) expressed in fetal erythroid precursors, which consequently increases the secretion of the neurogenic factor LIF. At the same time, inhibited neurogenesis was observed in mice on day 15.5 of pregnancy in response to the decreased LIF level after viral activation of the immune system [26].

No transplacental mother to fetus passage of proinflammatory cytokines such as TNFα, IL-1β, and IL-8 takes place during normal pregnancy. Their presence in the amniotic fluid indicates inflammation and an increased risk of premature delivery, embryonic mortality, cerebral palsy, and bronchopulmonary dysplasia in offspring (Figure 1) [27,28,29].

LPS-induced activation of the maternal immune system increases the synthesis of proinflammatory cytokines in the placenta. Thus, increased protein and mRNA levels of IL-1β, IL-6, TNFα were observed in the maternal peripheral blood, amniotic fluid, and placenta, six hours after LPS administration (100 µg/kg) on gestation day 18. The LPS-induced imbalance of maternal cytokines consequently triggered the secretion of the stress hormone ACTH in the fetal brain [30]. It is common knowledge that the ultimate stress effector molecules in the HPA are glucocorticoids (cortisol in human and corticosterone in rodents), which are involved in the programming of fetal development. In a normal pregnancy, they are inactivated in the placenta in syncytiotrophoblasts by the specific enzyme 11β-hydroxysteroid dehydrogenase type-2, thus restraining the stress response [31]. Suppressing the enzyme activity during stress will increase the level of glucocorticoids in the fetus [32]. Their high levels can affect HPA functions, behavior, as well as immune and neuroendocrine responses in offspring [33]. LPS downregulates the production of enzyme in late pregnancy, which disturbs neurite growth and formation of axonal connections [34].

The effects of LPS are realized not only via TLR4, but also via TLR2. TLR2 is believed to be involved in the recognition of a wide variety of infectious pathogens. In the third trimester of pregnancy, TLR2 is detected in the human placenta in endothelial cells and macrophages, comparably to its low level in the syncytiotrophoblast [35]. Conversely, high TLR4 levels are observed in the syncytiotrophoblast and preterm human placentas from complicated chorioamnionitis [36].

LPS administration to pregnant rats or mice modulates the synthesis of proinflammatory cytokines in both mother and fetus. An elevated expression of TNFα, IL-1β, and monocyte chemotactic protein (MCP-1) was revealed in the fetal brain [37]. Cai et al. [38] reported the top transcription of IL-1β in the fetal brain 1 h after an administration of high LPS doses (4 mg/kg) to rats on gestation day 18, while TNFα expression peaked after 4–24 h.

According to our data, a single administration of low LPS doses (45 µg/kg) to mice on gestation day 11.5 considerably increased the levels of IL-6, MCP-1, TNFα, and LIF in the maternal and fetal serum, amniotic fluid, and fetal spinal fluid after 1.5 h; these levels were maintained for 3–6 h and returned to the baseline [39].

The brain development relies on the balance between pro- and anti-inflammatory cytokines [40]. However, the role of anti-inflammatory cytokines in fetal development remains unclear. IL-10 is a key inhibitor of the synthesis of proinflammatory cytokines, and its fetal level depends on the LPS dose and method of administration to pregnant females. The level of IL-10 increases in the placenta and decreases in the fetal brain 6 h after intrauterine administration of 100 µg/kg LPS to pregnant rats [41]. IL-10 levels increased in the placenta and fetal liver and brain 1.5 h after intraperitoneal administration of 500 µg/kg LPS to pregnant mice [42]. An LPS-induced increase in fetal IL-10 expression decreases the risk of metabolic and behavioral disorders in offspring.

The transcription factor signal transducer and activator of transcription-3 (STAT3), which is activated by glycogen synthase kinase-3 (GSK3), is thought to be involved in the brain’s anti-inflammatory response to LPS, mediated by the IL-10 signal [43]. α-Melanostimulating hormone can also function as an anti-inflammatory cytokine. This neuropeptide suppresses the LPS-induced expression of TNFα, IL-1β, and IL-6 and modulates IL-10 synthesis [42].

## 5. Long-Term Effects of Perinatal Administration of LPS during Critical Periods of Development

In addition to cytokines, LPS induces the synthesis of the vascular endothelial growth factor (VEGF), antiapoptotic protein YB-1, and the neuronal differentiation factor, necdin, in the fetal mouse brain [37], as well as of the brain-derived neurotrophic factor in the fetal brain and nerve growth factor in the neonatal rat cortex [44].

The brain is the major LPS target (Figure 1). LPS increases the level of the glial fibrillary acidic protein (GFAP), an intermediate filament protein in astrocytes, the hippocampal and the cortex areas, and decreases the myelin level and microglial immunoreactivity in postnatal rats [38]. Intrauterine administration of LPS increases the levels of GFAP in the rat hippocampus on postnatal day 7, and in the cortex and corpus callosum on postnatal day 14 [45]. LPS-induced persistent inflammatory reactions in the rat brain lead to white matter injury in offspring on postnatal days 1 and 7. At the same time, an increased expression of GFAP was observed on postnatal days 1 and 3, and astrocytes actively proliferated and differentiated (astrogliosis) on day 7 [46]. In addition, high levels of proinflammatory cytokines TNFα, IL-1β, and IL-6 were observed in the maternal placenta as well as high levels of IL-1β in the fetal blood. The white matter injury increases the risk of cerebral palsy in newborns and in typical manifestations of schizophrenia in adulthood [24,45,47]. LPS administration to the male rats on days 15–16 of pregnancy also altered the morphology of pyramidal neurons in the prefrontal cortex and hippocampus during the earliest stages of postnatal development [48]. All these processes disturb the establishment of nervous connections, which can evoke long-term changes in psychomotor development, behavior, and neuroendocrine functions [49].

Prenatal exposure to LPS disturbs the dopaminergic and serotonergic systems. LPS administration to rats on gestation day 10.5 decreases the number of dopaminergic neurons in the substantia nigra as well as the number of serotonergic neurons in the raphe nucleus and hence, decreases the levels of dopamine and serotonin in the adults [2]. The progeny also demonstrated increased microglial activity and high levels of cytokines TNFα and IL-6 in the substantia nigra. LPS was proposed to suppress the release of glutathione, which has antioxidant properties—from glial cells [50], resulting in the death of dopaminergic neurons and eventually Parkinson’s disease [51].

LPS administered to mice on day 18 of pregnancy suppresses the expression of factors involved in neurogenesis, neuronal migration, and axonal cone growth, being semaphorin 5b and the transcriptional corepressor Groucho [37]. According to our data, LPS suppresses the initial stages of differentiation and migration of neurons producing gonadotropin-releasing hormone (GnRH). It is common knowledge that the reproductive function in mammals is governed by the HPG axis, which is established during perinatal development and up to the end of puberty. LPS reprograms HPG development during critical ontogenetic periods, which impairs the reproductive function and decreases fertility in adults [52,53,54].

Intraperitoneal administration of LPS (18 µg/kg) to the male rats on day 12 of embryonic development, when GnRH neurons are formed and start to migrate, retards their migration from the nasal region to the brain. The total number of GnRH neurons decreases by 50% in the forebrain of a 17-day-old fetus and by 17% in a 19-day-old fetus. At the same time, the number of neurons in the nasal region of 17- and 19-day old fetuses increased by 40% and 50%, respectively. LPS administration to pregnant females on day 15 of embryonic development had no effect on the total number of neurons, as well as on their distribution among the migration regions [55]. Decelerated intranasal migration of GnRH neurons was also observed after LPS administration (45 µg/kg) to mice on gestation day 11.5. IL-6 receptors and the intermediate filament protein, peripherin which are factors controlling the neuronal migration pathway, were revealed along the intranasal migration route of GnRH neurons on the olfactory and vomeronasal nerves growing into the brain [39]. Proinflammatory cytokines, among which IL-6 was most upregulated by prenatal LPS administration, can connect the maternal infection and the subsequently disturbed neuronal migration and differentiation in the fetus. The suppression of intranasal migration of GnRH neurons induced by LPS and their delayed appearance in the forebrain presumably alters the formation of essential axonal connections in the hypothalamus, which affects the key points of HPG development.

After maternal LPS exposure, adult male and female offspring decreased their hypothalamic levels of GnRH, blood levels of luteinizing hormone (LH) and sex steroids, as well as delayed puberty in rat females. At the same time, the level of sex hormones doubled in the prepubertal period relative to the control. The offspring of both sexes demonstrated decreased body weight (by 30%) and fertility [56].

Individual sexual maturation depends on the interplay of internal and external factors: Body weight and fat content, concentrations of sex steroids, and advancement of the neuroendocrine mechanisms controlling GnRH neuronal activity in the brain [57,58]. Gonadal development, steroidogenesis, and functions in males and females is modulated by cytokines IL-1β, TNFα, IL-6, IL-8, MCP-1, and LIF [59,60]. Urogenital ridge cells express LIF receptors [61]. IL-8 and MCP-1 are involved in follicular development and atresia, ovulation, and corpus luteum function in females. IL-6 induces testicular resistance to LH and suppresses steroidogenesis by Leydig cells, thus modulating the male reproductive function. IL-6-induced production of LH and IL-1β activates the proteolytic enzymes involved in the follicular atresia in females [62]. IL-6 receptors were revealed in human ovarian cells [63]. TNFα and IL-1β suppress steroidogenesis in undifferentiated ovarian cells and stimulate progesterone synthesis in differentiated ones [59].

According to our data, apart from the disturbed migration of GnRH neurons, prenatal exposure to LPS affects testicular and ovarian development in rats, which can be attributed to the high levels of proinflammatory cytokines in the fetus, and sex steroids in the prepubertal period. Antagonists of receptors for testosterone (flutamide) and estradiol (fulvestrant) administered in the period of elevated hormones to females and males, respectively, normalized sexual maturation in adult offspring. The latter almost completely restored their body weight, sexual gland structure, sex hormone levels, and reproductive capacity [64,65] (Figure 2).

## 6. Conclusions

The analysis of published and original data demonstrates that LPS-induced activation of the maternal immune system in early pregnancy impacts on the formation of the HBA and HPG, as well as their functionality in adults. Early development is the period in which the epigenetic mechanisms providing for the adaptive plasticity of these systems are realized. Compromised molecular mechanisms controlling development during this period can induce long-term or irreversible changes in their functions. However, the processes of establishment of these systems are not strictly genetically determined. They feature functional lability and are responsive to many regulatory factors, which makes it possible to correct the abnormal development.

Recently, efforts have been made to correct the negative impact of LPS on the development of the fetal brain. Maternal administration of the antioxidant N-acetyl cysteine or zinc salts prior to LPS injection inhibited the production of cytokines TNFα, IL-6, and IL-10 as well as local inflammation in the fetal brain, which abolished the long-term negative consequences of inflammation [66,67]. A similar effect was also observed after administration of magnesium sulfate, which decreases the LPS-upregulated levels of phospho-nNOS, NF-κB, CCL2, and proinflammatory cytokines. The effect of magnesium sulfate was mediated by the activation of N-methyl-D-aspartate receptors [49,68]. A mitigation of the negative consequences of the maternal immune system activation by LPS or double-stranded RNA (poly I:C) was induced by immediate administration of vitamin D [69], polyunsaturated fatty acids (PUFAs) [70], interferon (IFN)γ, and antibodies against cytokines IL-6 (IgG1-antibodies), IFNγ (IgG2a-antibodies), and IL-1β (IgG1-antibodies) [71].

In the current study we focused on the prenatal effect of LPS on the development and functioning of the GnRH-producing and reproductive systems. The revealed developmental abnormalities of these systems are partially due to the increased levels of proinflammatory cytokines in the fetus and the consequently increased levels of sex steroids in the prepubertal period (Figure 2). In this context, two periods can be recognized for the correction of negative LPS effects: immediately before, after, or several hours after LPS administration prenatally or in the period of elevated sex hormones prepubertaly. Our correction of reproductive abnormalities by antagonists of sex steroids restored the reproductive capacity of the progeny. Thus, reproductive abnormalities in adults should be controlled and prevented in both perinatal and infantile periods.

## Figures and Tables

**Figure 1 ijms-19-03695-f001:**
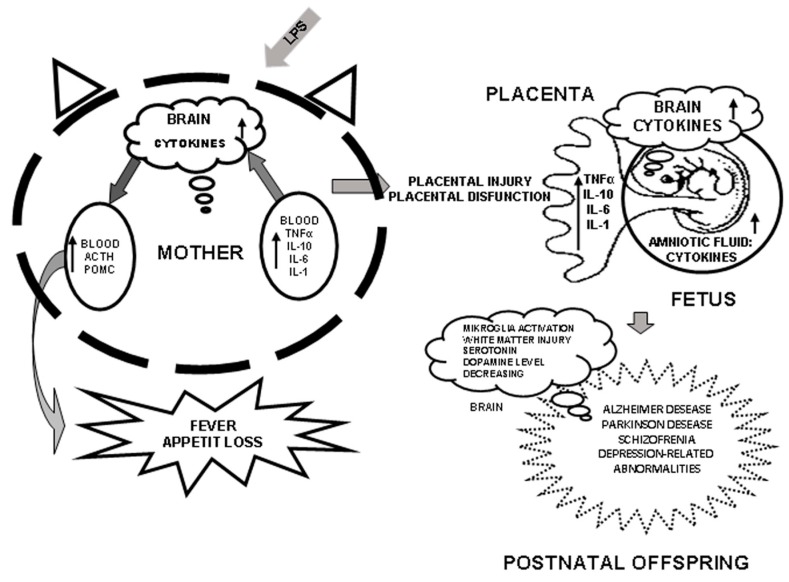
Effect of proinflammatory cytokines on fetal brain development after prenatal exposure to LPS.

**Figure 2 ijms-19-03695-f002:**
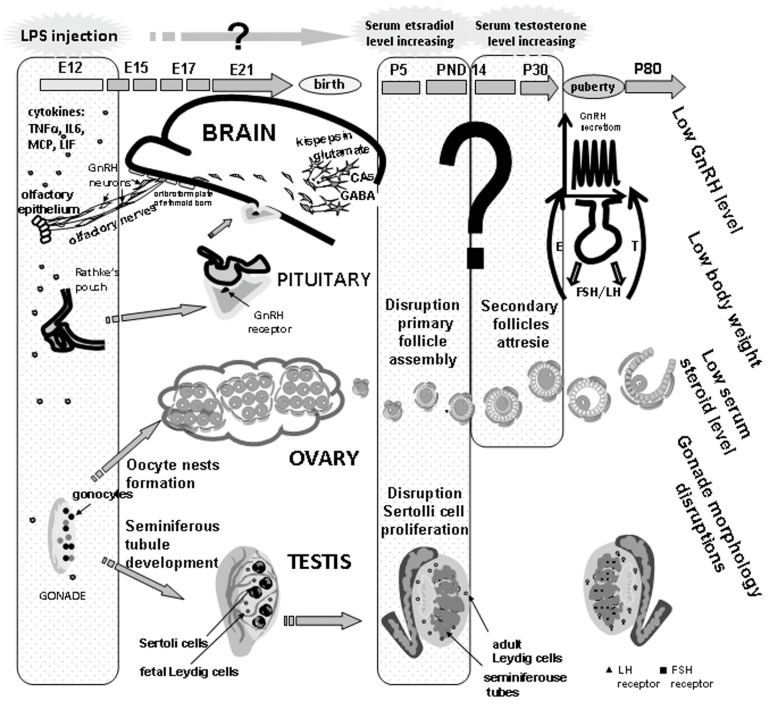
Mechanisms underlying the developmental origins of female and male postnatal sexual abnormalities after prenatal exposure to LPS.

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
