# Peer review of "Prenatal Programming of Neuroendocrine System Development by Lipopolysaccharide: Long-Term Effects"

_ijms, 2018, doi:10.3390/ijms19113695_

Reviewer 1 Report

LPS is one of the significant risk factor for infection by bacteria that cause inflammation. Izvolskaia et al., reviewed about LPS from it's component to the mechanism that disturb the pregnant. Though the manuscript is well designed and written there are few comments regarding the manuscript:

1) Authors used many abbreviations, but some of them were not explained, especially in Abstruct part. Authors should explain all of them in the text.

 2) In Line 128, authors mentioned 'As mentioned above, LPS effects are realized via Toll-like receptors including TLR2 and TLR4.' However, the authors don't mentioned about TLR2 signaling in above sentences. Generally TLR2 is stimulated by gram positive bacteria, authors should explain that how TLR2 signaling is required for LPS-induced TLR4 signaling.

 3) Systemic immunity in pregnant woman are more tolerance to avoid a miscarriage. Thus, produciton levels of IL-10 in pregnant women is also higher than non pregnant woman? The levels of regulatory cytokine productions are depend on only fetal productions in pregnant woman by LPS stimulation? 

 4) How is regulatory T cells and M2 macrophages that produce mainly IL-10 during pregnant by LPS stimulation?

 5) What a spontaneous source of LPS infection during pregnant? What clinical case the authors represent by Intraperitoneal administration of LPS?

 line 79, granulocyte-macrophage colony-stimulating factor (GM-CSF)

 line 91, Mitogen-activated protein kinase phosphatase (MKP)-1

 line150, STAT3 (signal transducer and activator of transcription-3) ->

 signal transducer and activator of transcription-3 (STAT3)

 line 151, 156, authour don't give abbr. to 'glycogen synthase kinase-3' or 'vascular endothelial growth factor'. However, GSK3 or VEGF might be more general than original.

Author Response

1) Corrected. Line 18 and 21.

2) In line 135-136.  LPS effects are realized not only via TLR4, but also via TLR2. TLR2 is believed to be involved in the recognition of a wide variety of infectious pathogens.

Previous studies have suggested that TLRs, activated at the maternal–fetal interface, are responsible for the response to pathogens and regulation of infection-related inflammation during pregnancy. TLR2 is an essential pattern recognition receptor for MALP-2 (mycoplasmal lipopeptides (macrophage-activating lipopeptides, 2 kDa), just as TLR4 is for LPS. An adaptor molecule, MyD88, is involved in the signaling pathway via TLR4. Upon stimulation, MyD88, which binds to TLR4, recruits IRAK to the receptor. IRAK then activates TRAF6, leading to activation of NF-kB and JNK. Indeed, both MyD88- and TRAF6-deficient mice displayed hyporesponsiveness to LPS. Both peptidoglycan and MALP-2 have been shown to be recognized by TLR2. Thus, MyD88 is the adaptor molecule shared by TLR2- and TLR4-mediated signaling pathways. The mycoplasmal lipopeptide MALP-2 is structurally related to bacterial lipopeptides and lipoprotein, all of which have been shown recognized by TLR2.

The existing literature has illustrated the importance of TLR2 and -4, which are involved in the innate immunity of the fallopian tube, with the most extensive expression in the tube along the female reproductive tract during the nonpregnant state. However, there is no information on TLRs in the fallopian tube with tubal pregnancy. In addition, the expression of TLR2 and TLR4 are influenced by the estrous cycle. So the expression of TLR2 and 4 in different segment of genital tracts have different response after LPS-stimulation.

Ji, Y.F.; Xu, J.; Zhang, T.; Chen, L.Y. Decreased Toll-like receptor-2 messenger ribonucleic acid and increased Toll-like receptor-4 in the tubal epithelium next to the infiltrated trophoblasts during tubal pregnancy. Fertil Steril. 2017, 107, 282–288.

Silva, A.L.; Fry, W.H.; Sweeney, C.; Trainor, B.C. Effects of photoperiod and experience on aggressive behavior in female California mice. Behav Brain Res. 2010, 208, 528–534.

3)Several authors have suggested that the maternal innate immune system may be more active in pregnancy to compensate for a possible repression of adaptive immune system and that this overactivity may be responsible for pregnancy complications including preterm labor  and preeclampsia [ Tang et al., 2015 ]. Few studies have investigated differences in the peripheral blood mononuclear cells responses in pregnant and non pregnant  individuals and found that the cytokine (interleukin (IL)–6, IL-10, and IL-17) response was increased in the cells from pregnant women exposed to a variety of agents (umbilical cord blood (UCB) of the mother’s own child, third-party UCB, phytohemagglutinin, and anti-CD3 antibody)

Similarly, monocyte-derived dendritic cells from pregnant women exposed to cytokines (IL-1. and tumour necrosis factor alpha (TNF.)) showed a marked increase in IL-10, and a similar trend in response to LPS. In study of Zollner, 2017 was marked the increase in the most cytokines/chemokines in both pregnant and nonpregnant mice, only the response of the anti-inflammatory cytokine, IL-4, was greater in pregnant mice. This is in contrast to the only previous study to compare the response to LPS in NP and pregnant mice, in which the increase in IL-6, IFN, and TNF was greater and of IL-10 was reduced in pregnant mice; these changes were associated with a greater mortality in the pregnant mice. These data are important as they show that the response of the innate immune system to LPS is similar in pregnant and nonpregnant.

Zöllner, J.; Howe, L.G.; Edey, L.F.; O'Dea, K.P.; Takata, M.; Gordon, F.; Leiper, J.; Johnson, M.R. The response of the innate immune and cardiovascular systems to LPS in pregnant and nonpregnant mice. Biol Reprod. 2017, 97, 258–272.

4) During pregnancy, the balance of Th1 (cell-mediated immunity) and Th2 (humoral immunity) cytokines is characterized by an initial prevalence of Th2 cytokines, followed by a progressive shift toward Th1 predominance late in gestation, that when is abnormal, may initiate and intensify the cascade of inflammatory cytokine production involved in adverse pregnancy outcomes.

T regulatory (Treg) cells are a recently discovered subset of T-lymphocytes with potent suppressive activity and pivotal roles in curtailing destructive immune responses and preventing autoimmune disease. Treg cells are increased in the blood, decidual tissue and lymph nodes draining the uterus in pregnancy. Dendritic cells controlling Treg cell populations can be targeted in vivo to enhance Treg cell numbers and function.

According Guerin, 2009 studies in mouse models show that Treg cells are essential for maternal tolerance of the conceptus, and that expansion of the Treg cell pool through antigen-specific and antigen non-specific pathways allows their suppressive actions to be exerted in the critical peri implantation phase of pregnancy. In women, Treg cells accumulate in the decidua and are elevated in maternal blood from early in the first trimester. Inadequate numbers of Treg cells or their functional deficiency are linked with infertility, miscarriage and pre-eclampsia. During pregnancy, the balance of Th1 (cell-mediated immunity) and Th2 (humoral immunity) cytokines is characterized by an initial prevalence of Th2 cytokines, followed by a progressive shift toward Th1 predominance late in gestation, that when is abnormal, may initiate and intensify the cascade of inflammatory cytokine production involved in adverse pregnancy outcomes.

Guerin LR, Prins JR, Robertson SA. Regulatory T-cells and immune tolerance in pregnancy: a new target for infertility treatment?Hum Reprod Update. 2009 Sep-Oct;15(5):517-35. doi: 10.1093/humupd/dmp004. Epub 2009 Mar 11. Review.

5) Peripheral administration of LPS from gram-negative bacteria is a well-characterized model of inflammation in rodents [6]. A single intraperitoneal injection of low doses (50-100 µg/kg) of 125I-labeled LPS to pregnant mice has revealed no LPS in fetal tissues [24], while radioactive signal was detected in both placental and fetal tissues after intravenous injection of a higher dose (up to 1000 µg/kg) [1].

LPS was used in the 50s of the last century to simulate all stages of the inflammatory process without infecting the body with potentially dangerous bacterial infections. LPS administered intravenously or intraperitoneal accumulates primarily in the liver as well as in the spleen, blood vessel walls and pulmonary alveoli. In macrophages, LPS in combination with auxiliary receptor molecules CD14 and MD2 triggers the transmission of a "danger signal" via Toll-like receptors (TLR4). Thus, the injections of LPS imitates the general inflammatory process that occurs in the body (pregnant or not), which in turn affects various neuroendocrine processes both in the adult animal and developing fetuses.

line 79, granulocyte-macrophage colony-stimulating factor (GM-CSF) - corrected

line 91, Mitogen-activated protein kinase phosphatase (MKP)-1 - corrected

line150, STAT3 (signal transducer and activator of transcription-3) -> corrected

signal transducer and activator of transcription-3 (STAT3)

line 151, 156, authour don't give abbr. to 'glycogen synthase kinase-3' or 'vascular endothelial growth factor'. However, GSK3 or VEGF might be more general than original

Reviewer 2 Report

This is a very nicely written review article highlighting the effect of lipopolysaccharides (LPS) induced inflammation in mothers and infants during pregnancy. The authors have highlighted that LPS mediated immunopathogenesis leads to several pregnancy-related complications and developmental defects in the offsprings.  

I have only minor comments:

1) Line 11: The statement " Significant risk..." is confusing. Please modify it to convey the message clearly.

2) Line 36: "....to stimulate bacterial infection..." does author means mimic bacterial infection, as LPS inoculation cannot be considered an infection.

Author Response

1) Significant risk factors programming fetal development include prenatal stress induced by bacterial infection simulated by lipopolysaccharide (LPS) in the experiment.

Corrected  Maternal immune stress by bacterial infection simulated by lipopolysaccharide (LPS) in the experiment is considered as a powerful programming factor of fetal development.

2) Yes,  thank you. Corrected.